# IS THIS A REAL IMAGE?

## ABSTRACT

With the rapid development of generative AI and LLMs in recent years, a challenging issue has emerged: how can we determine whether an image or a video on the internet is real or AI-generated? AI-generated fake images and videos pose various potential risks, such as fraud, fake news, and copyright issues. We focus on the challenges generative AI brings to internet regulation and will discuss how we can address these challenges by using asymmetric encryption and trusting chains to sign image and video files digitally. In this way, anyone can verify the authenticity of a given image or video.

## 1 CHALLENGES

Generative AI and LLMs Vaswani et al. (2017); Goodfellow et al. (2014); Yang et al. (2023) can be considered one of the most significant technological breakthroughs in the field of AI and even in the entire tech industry in recent years. However, as generative AI technology rapidly advances, a challenging question has arisen: how can we determine whether an image or a video on the internet is real or AI-generated? As shown in Figure 1, there is an account on Chinese TikTok that posts AI-generated videos and images of national leaders (for privacy reasons, we have temporarily hidden the account ID). Currently, the fake contents posted by this account is just used for entertainment, and due to technical limitations, most of them can still be easily recognized as fake contents by human eye. However, as technology and hardware continue to evolve, the barrier of using generative AI will decrease, and the content it produces could become increasingly difficult for humans to visually recognize whether it is real or AI-generated.

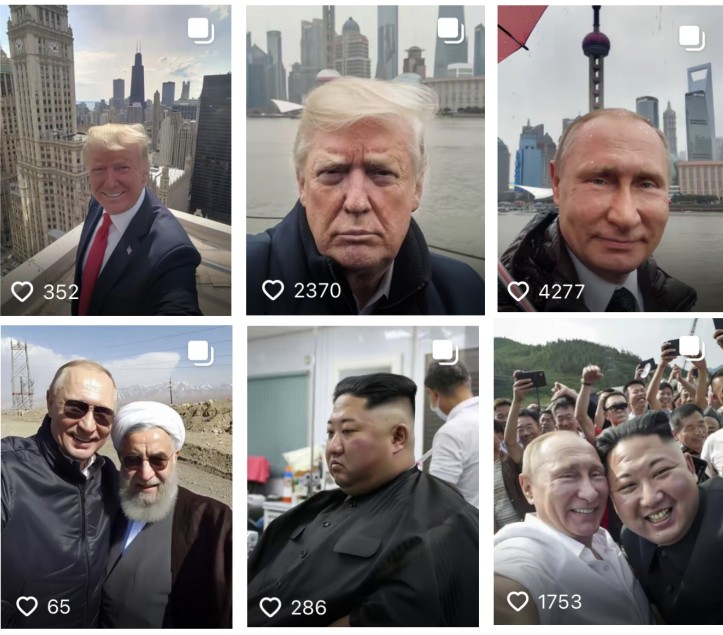

Figure 1: All the images are AI-generated fake images.

Consider the following scenario: a major financial institution intends to short Tesla stock. They begin by using generative AI technology to create a fake video of Elon Musk, in which he expresses his view that Tesla's current stock price is overvalued. This institution then posts this fake video on the internet, and within just a few minutes, the impact of this fake video could be reflected on Tesla's stock price. By the time Elon Musk himself steps forward to clarify that the video is fake, the influence on the stock price has already been done.

If you think the above example is far away from the daily lives of ordinary people, generative AI technology could also be used by malicious individuals in our everyday lives for activities such as fraud. For example, AI face-swapping (also known as deepfake Mirsky & Lee (2021)) can easily replace a person's face with someone else's in any video. If someone were to take a girl's photo from her social media and use AI to swap her face onto explicit videos, then threaten her and her family with these videos, how could we prove these videos are fake, rather than allowing such incidents to happen?

## 2 GOVERNANCE OF AI TECHNOLOGY

Facing the challenging issues posed by generative AI, current solutions primarily focus on two aspects: public education and internet regulation. Firstly, there is still a significant gap in educating the general public about AI. Many people are unaware of the progress that generative AI technology has made, and many people do not believe that AI can produce such realistic fake images and videos. They are even less likely to believe that current AI technology can be used for malicious purposes, such as fraud, fake news, copyright infringement, and extortion.

In addition to the popularization of AI education, internet regulation is also one of the means to address current issues. For example, many social network Apps require users to indicate whether the images or videos they upload are generated by AI. Furthermore, some Apps implement a manual review process: if an image or a video is identified by the administrators as AI-generated, it will be labeled with a message: *"This content may be generated by AI, please pay attention in identification."*

However, the above solutions do not fundamentally address the issue generative AI brings to internet regulation, as a thief would not label himself with the words: *"I am a thief."* Similarly, if someone intentionally uses generative AI technology for malicious purposes, they will undoubtedly find ways to circumvent internet regulation. Therefore, our question is: Is there a method that can fundamentally determine, through technological means, whether an image or a video is real or generated by AI?

The current explorations in academia focus on training a detection model using machine learning methods to determine whether an image is real or is AI-generated Suganthi et al. (2022); Hsu et al. (2020). However, machine learning is ultimately a probabilistic approach, and its accuracy is still far from meeting the requirements of practical applications.

## 3 SELF-VERIFICATION OF SOURCE

Whether it is an image or a video, it must either be captured by a device, such as a camera or a smartphone, or generated by a generative AI program, such as Midjourney. Therefore, as long as the image or video can prove that it was captured by a real device rather than generated by AI, its authenticity can be verified. We define this process as self-verification of source.

As shown in Figure 2, Bob takes a photo of his pet dog with his smartphone and uploads it to his social network. How can we prove that this image is a real dog captured by a smartphone, rather than generated by an AI program? An intuitive idea is that when the smartphone takes this photo, it simultaneously writes a message in the metadata of the image: *"This image was captured by the xxx smartphone."* Similarly, if the image were generated by a generative AI program, the AI program needs to write a message to the metadata: *"This image was generated by the xxx generative AI program."* Then, when the image is uploaded to a social network App, the app automatically verifies the metadata source information, thereby determining whether the image is real or is AI-generated. However, as we mentioned, a thief would not label himself with the words: *"I am a thief."* If someone intends to use generative AI for malicious purposes, they would not label the

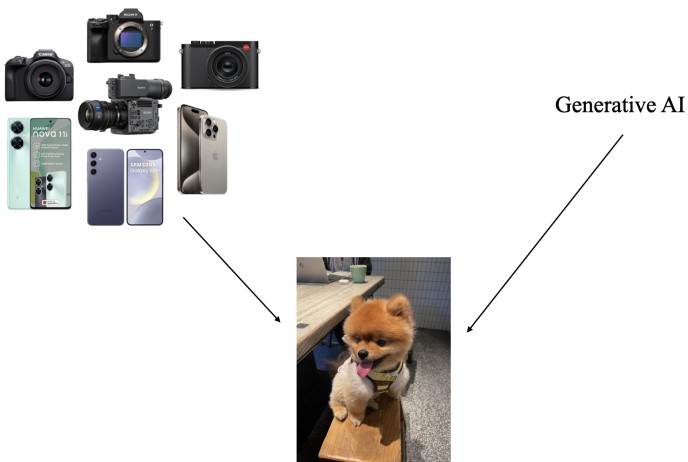

Figure 2: Two ways to get a photo of your pet dog.

metadata as an AI-generated fake image; instead, they would copy the metadata of a real device to bypass the App's verification process.

Existing digital watermarking techniques Mohanty (1999); Podilchuk & Delp (2001) can address some of the aforementioned issues. However, a major issue with digital watermarking is that it alters the original data of the image or video, and it may become ineffective during the transmission of the file due to secondary modifications. For example, if you import an AI-generated image into Instagram for filter adjustment, the digital watermarking will be modified. In the next few sections, we will discuss how to solve this problem using asymmetric encryption algorithm and trusting chains.

## 4 ASYMMETRIC ENCRYPTION

Information encryption is a well-established field in cryptography, and asymmetric encryption algorithms Simmons (1979) have been widely applied in information communication and security. Here, we provide a brief introduction to the core idea of asymmetric encryption. Suppose Bob wants to send a message to Alice, but there is a risk that the message could be intercepted during transmission. Therefore, Bob needs to encrypt the plaintext message, and once Alice receives the encrypted ciphertext, she can decrypt it locally. In asymmetric encryption, both Bob and Alice each have two keys, known as public key and private key, which are two different strings. The private key is a string known only to the holder of the key, while the public key is visible to everyone. As shown in Figure 3 (a), before sending a message to Alice, Bob uses Alice's public key to encrypt it, and upon receiving the encrypted ciphertext, Alice decrypts it using her private key. Throughout the process, since only Alice knows her private key, no one else can decrypt the ciphertext.

The asymmetric encryption algorithm described above uses Alice's public and private keys. In addition to using Alice's keys, the asymmetric encryption algorithm can also use Bob's keys. As shown in Figure 3 (b), before sending a message, Bob encrypts it using his private key, and when Alice receives the encrypted message, she decrypts it using Bob's public key. Since Bob's public key is visible to everyone, this process does not actually encrypt the information; rather, it creates a digital signature. In practical applications, Bob first performs a hash operation on the message he wants to send, generating a message digest. Bob then encrypts this hashed digest using his private key and sends the encrypted digest along with the message to Alice. Upon receiving the message, Alice first decrypts the digest using Bob's public key and then performs a hash operation on the original message. She then compares the hashed message with the decrypted digest. If the two match, it confirms that the message was indeed sent by Bob. This process is also known as digital signature.

Back to the issue of self-verification of source for an image. By leveraging the concept of digital signature, if a capturing device can automatically generate a digital signature for the image after

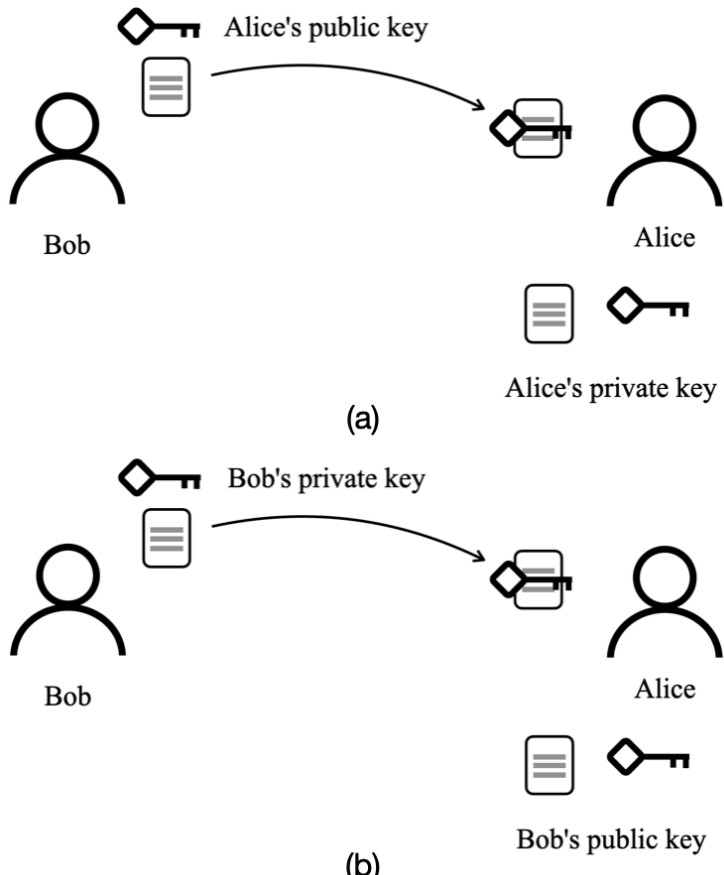

Figure 3: Encrypting and decrypting information using asymmetric encryption algorithm.

capturing it and then embed this signature in the image's metadata, it would confirm that the image indeed comes from a real capturing device rather than being a fake image generated by an AI program.

## 5 MULTI-SIGNATURES

Real-life scenarios may be more complex than the example of Bob taking a photo of his pet dog. For instance, Bob might use a Canon camera to take a landscape photo and then further process the image in Adobe Photoshop, such as adjusting its size and colors. Similarly, Bob could use his iPhone to record multiple video clips and then edit them together using Apple iMovie. Whether it's image or video post-processing, such modifications change the original data, which would invalidate the initial digital signature from the capturing device. To address this issue, third-party editing software could use the same algorithm to apply a new digital signature to the modified image or video.

As shown in Figure 4, after Bob takes a photo with a Canon camera in RAW format, the camera uses its private key to sign a digital signature to the photo, proving that it is a real image. Bob then imports this photo into Adobe Photoshop for further editing. At this stage, Adobe software first verifies the photo's digital signature to ensure that it is real. After Bob completes the edits, Adobe uses its own private key to sign a second signature to the exported image. If Adobe cannot confirm that the photo comes from a real capturing device during the initial verification, it will not sign a second signature. In real life, there may be third, fourth, or even more signatures, all following the same verification process. For example, after editing the photo in Adobe Photoshop, Bob imports this photo into Instagram for filter adjustment. Instagram will also verify Adobe's signature and then sign a third signature. After that, Bob sends the photo to a friend via WeChat. When WeChat sends

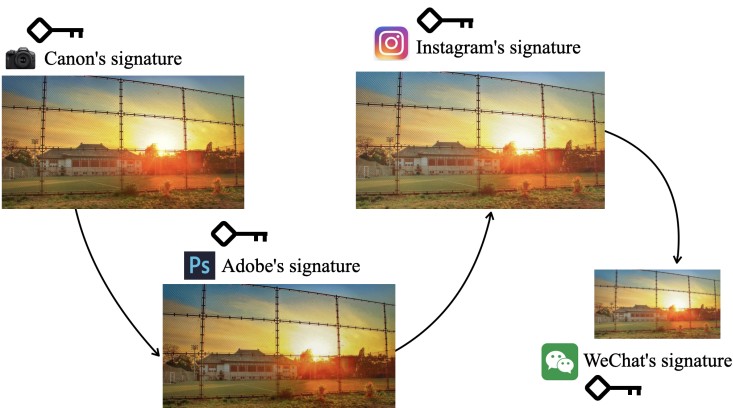

Figure 4: Multi-signatures with four entities.

the image, it compresses it first, which modifies the original data and invalidates Instagram's prior signature, requiring WeChat to use its own key to re-sign the image.

# 6 TRUSTING CHAINS

As of now, our method can effectively apply a digital signature to an image or a video to verify whether it is real or AI-generated. However, this approach requires a fundamental condition: trust in the signer. For example, if a Canon camera uses its private key to sign an image, we can verify the image using Canon's public key. This process is only reliable if we trust Canon — we trust that Canon will not use its private key to sign an image that we don't know where it comes from and that Canon can securely manage its private key, ensuring it is not leaked to unauthorized parties.

Similarly, in a multi-signatures process, if Bob imports a photo taken by a Canon camera into Adobe Photoshop for further editing, Adobe Photoshop would first verify it and then re-sign the image provided by Canon. This process is reliable because we trust Adobe as a company — we trust that Adobe has indeed verified the image's original source and that Adobe securely manages its own private key. However, if the editing software is not Adobe Photoshop but an unauthorized third-party App, we cannot be assured that this third-party App would act in good faith. For example, this third-party software could probably provide an authenticity signature for a fake photo generated by AI.

As shown in Figure 5, (Canon - Adobe PS - Instagram - WeChat) forms a trusted chain with a length of 4. This chain is considered trustworthy because we assume these four entities are reliable by default. However, if we introduce an unauthorized third-party App at any point along this chain, the trust chain becomes untrustworthy from the moment that third-party App is added. In real life, we cannot ensure that an image or a video won't be edited by an unauthorized third-party App. To address this issue, we use a conservative solution: as long as an untrusted third-party entity (which could be either software or hardware) is encountered in the trusting chain, the subsequent entities will no longer verify the authenticity of the image and will not sign it anymore. This status will persist until a new trusted entity signs the authenticity of the image (for example, when the image is uploaded to TikTok and triggers manual review and verification). The advantage of this approach is that it ensures that any signed image is authentic, provided that certain large institutions are universally recognized as trusted entities. Examples of such entities include camera manufacturers, smartphone manufacturers, major social media platforms, and operating systems. Using this solution, it is necessary for the software and hardware communities to establish mechanisms that allow new third-party entities to obtain trusted authorization.

However, a major concern of this solution is that it might be criticized as being overly conservative. We will discuss this issue in two scenarios:

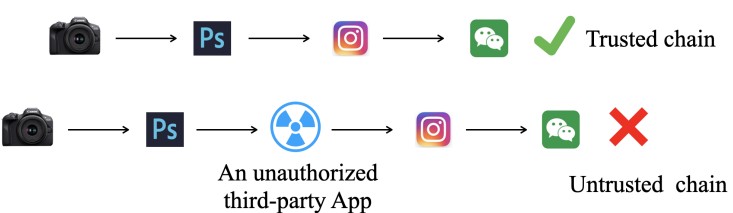

Figure 5: Trusted chain and untrusted chain

(i) If generative AI technology is abused in the future and the internet is flooded with malicious fake content, then it becomes essential for an image to obtain an authentic signature. Therefore, our conservative solution will be the right choice.

(ii) If, in the future, we find that our previous discussions about generative AI development were overly pessimistic and that almost no one uses generative AI technology for malicious purposes, then the importance of an image or a video having a signature decreases. Therefore, even if an image does not have an authenticity signature, it does not affect its actual circulation and use.

## 7  AUTHORIZATION MANAGEMENT

So far, our entire system is based on a foundational assumption: we trust a few existing large signing entities whose public keys are visible to everyone. However, new software and hardware will inevitably emerge, and how to grant signing authority to these new entities is a critical issue we need to address. To tackle this, we suggest a decentralized voting mechanism to manage the signing entities. For example, a new entity seeking signing authorization must obtain over 90% of the votes from the entire community. Also, we suggest using hierarchical management for these entities. For example, if a new app receives only 80% of the votes, it can still be granted signing authorization, but its trust level will be lower than that of the existing large signing entities in the community, as shown in Figure 6.

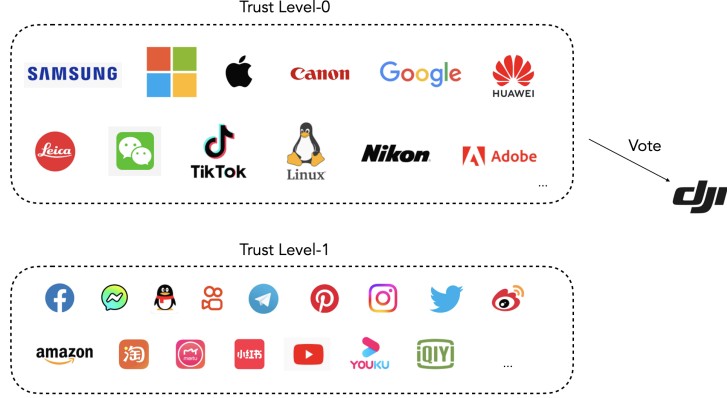

Figure 6: A new entity wants to obtain signing authorization.

High-level trust entities will only sign entities that are at the same level or higher than themselves. Using trust level for signing entities allows us to verify the authenticity of images with greater fine-grained management. For instance, critical content such as speeches by national leaders must obtain the highest level of verification. On the other hand, for entertainment-related content, whether it requires the highest level of verification depends on the review standards of different apps.

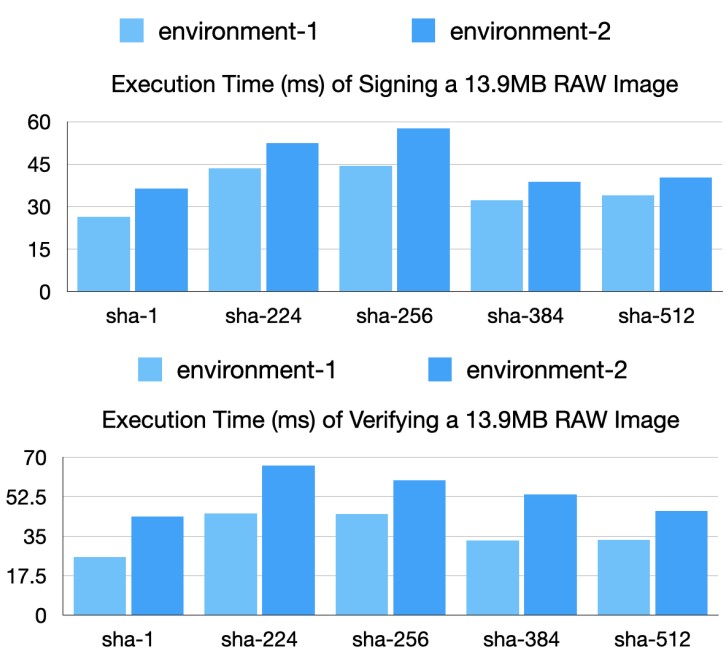

Figure 7: Execution time of signing and verifying a high-quality lossless format image.

## 8 IMPLEMENTATION AND EXPERIMENT

The widely used asymmetric encryption algorithms include the RSA algorithm Boneh et al. (1999) and the EC (Elliptic Curve) algorithm Koblitz (1987). Implementing an efficient RSA or EC algorithm that can run stably in an industry environment is not an easy task. However, with the help of existing tools such as OpenSSL, we can conveniently implement the aforementioned digital signature algorithm. We developed a small tool called libsig and our code is open-sourced at: https://github.com/chao0x11/libsig. In libsig, we choose EC algorithm as our encryption and decryption algorithm. Particularly, we choose *spec251k1* as our elliptic curve, which has been verified its security in many systems such as Bitcoin Nakamoto (2008). We choose the secure hashing algorithm (SHA) NIST (2015) as our hashing algorithm. The secure hashing algorithm has also been widely verified for its security in preventing hashing collisions.

In practical applications, runtime overhead is one key factor that we need to focus on. This is because our method could be deployed in various environments, some of which have limited computational resources, such as small embedded devices. We tested our signing and verifying method on a high-quality lossless image using different hashing algorithms, and the experimental results are shown in Figure 7.

We selected two experimental environments: a Cloud Linux VM (environment-1) and a MacBook Pro (environment-2). We found that in both environments, the signing and verifying latency for a 13.9MB high-quality lossless image can be completed in a blink. This indicates that our method can complete the signing and verifying process for most image files within 1-2 seconds even in environments with computational power an order of magnitude smaller than our experimental setups. For example, in certain real-time scenarios, devices need to capture and upload images or videos simultaneously. If the hashing and encryption algorithms take too much time, the usability of our method in these scenarios would be significantly reduced.

We also tested the scalability of our signing and verifying method on environment-2, and the experimental results are shown in Figure 8. In this experiment, we choose sha-256 as our hashing algorithm. Our experiments show that the overhead involved in signing and verifying large media files is non-negligible. An optimization we can apply here is to sign the data using sampling meth-

ods, rather than signing the entire data. The detailed information of our experimental environments are as follows:

Environment-1: Ubuntu 22.04.5 LTS, 2.5 GHz Intel(R) Xeon(R) Platinum, 2vCPU, 3GB memory.

Environment-2: Mac OSX (12.7.6), 3.1 GHz Intel(R) dual-core i5 process, 8GB 2133 MHz LPDDR3 memory.

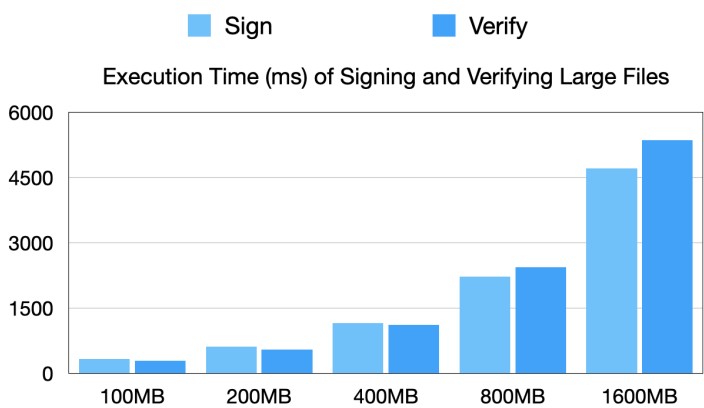

Figure 8: As the size of the file grows, the execution time of singing and verifying grows linearly.

## 9 FUTURE EXPLORATION

With the further development of generative AI technology, we will see an increasing amount of AI-generated contents on the internet in the future. To clarify, this paper does not seek to block the free development of AI. On the contrary, we sincerely hope that AI technology can continue to develop in the right direction. Furthermore, in the future, we may encounter many more complex real-world cases than those mentioned in this paper. For instance, if a video contains both real content and AI-generated content, how should we sign for the video? Additionally, how can a third-party App or device obtain the authorization to sign a signature? How can software and enterprises improve key management? Moreover, we may also face a small number of individuals who will go to great lengths to use generative AI for malicious purposes. Facing the various unforeseen situations that may arise in the future, our philosophy and principle are as follows:

*It is impossible for the world to be completely free of evil-doing, what we need to do is to increase the cost of evil-doing as much as possible.*

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
