# OpenReview forum: "Is this a real image?"
_ICLR.cc/2025/Workshop/BuildingTrust — Submitted to BuildingTrust_

### Official Review · Reviewer_3AQF · 2025-02-16
**Review of the paper**

**Rating:** 7
**Confidence:** 3

**Review:**

## Summary

This paper discusses the challenges posed by generative AI technology, particularly in determining whether images or videos on the internet are real or AI-generated. The paper points out that solutions to this issue include raising public awareness of AI technology and strengthening internet regulation, while also proposing specific technical solutions.

The paper suggests using digital signature technology to verify the authenticity of images or videos. These digital signatures can confirm whether an image came from a real device (such as a smartphone or camera) rather than being generated by AI. To ensure the integrity of these digital signatures, the paper recommends using asymmetric encryption algorithms to encrypt and decrypt messages and introduces the concept of trust chains to ensure the reliability of the digital signatures.

However, there are challenges to these methods. The paper mentions the issue of maintaining the reliability of digital signatures when third-party applications perform editing. Furthermore, how to handle the authorization of emerging technologies and ensure secure key management need further exploration.

In summary, this paper provides an in-depth analysis of the issues surrounding generative AI technology and proposes concrete solutions, particularly in terms of digital signatures and trust chains, while also highlighting potential challenges and difficulties in implementing these technologies.

## Strengths

**1. The topic is crucial**: The issue if detecting AI-generating images has became more and more important due to the rapid development of technology. The paper proveides a system makes it easier to tell if an image or video is real or AI-generated.

**2. Keep the balance between technology and ethics**: The paper presents a very practical and ethically grounded viewpoint: increasing the cost of wrongdoing rather than trying to eliminate it entirely. This reflects a responsible attitude towards the development of AI technology and connects technological advancement with societal ethical concerns.

**3. Proposes specific, feasible solutions**: The paper not only analyzes the problems but also proposes solutions like digital signatures and trust chains, and discusses how these solutions can be implemented within existing technological frameworks, offering concrete guidance for practical applications.

## Weaknesses

**1. Overly conservative approach to third-party applications**: While the article suggests a conservative solution, such as introducing third-party applications making the trust chain unreliable, this may be overly cautious and limit some innovative and emerging applications, potentially hindering the progress and use of new technologies.

**2. High implementation difficulty**: The proposed digital signature systems and trust chain technologies, although theoretically feasible, would require significant resources and collaboration to implement in practice. Promoting new technological standards globally could face numerous challenges.

---

### Official Review · Reviewer_R62E · 2025-02-27
**Review of "Is This a Real Image?" - Misalignment with workshops goals.**

**Rating:** 3
**Confidence:** 4

**Review:**

The paper addresses a critical and timely challenge—identifying AI-generated images and videos to mitigate risks such as misinformation, fraud, and copyright violations. The authors propose a cryptographic-based verification system that leverages **asymmetric encryption, digital signatures, and trust chains** to establish the authenticity of media files.

While the problem itself is important, the proposed solution **does not incorporate Artificial Intelligence (AI) or Machine Learning (ML)**. Instead, it relies solely on cryptographic techniques, which, while effective for ensuring file integrity, do not align with the workshop’s focus. Furthermore, beyond the discussion on **trust chains** for implementing their framework, the paper does not introduce any significant novelty in the task of detecting AI-generated images.

Given the **lack of novelty** and **misalignment with the workshop’s objectives**, I am inclined to recommend **rejecting** this paper.

---

### Decision · Program_Chairs · 2025-03-04

**Decision:**

Reject

**Comment:**

While the problem is important, this paper does not provide a novel solution.